# Continuous control of classical-quantum crossover by external high pressure in the coupled chain compound CsCuCl₃

Daisuke Yamamoto [1,2✉], Takahiro Sakurai [3✉], Ryosuke Okuto[4], Susumu Okubo[5], Hitoshi Ohta [5], Hidekazu Tanaka[6] & Yoshiya Uwatoko [7]

In solid materials, the parameters relevant to quantum effects, such as the spin quantum number, are basically determined and fixed at the chemical synthesis, which makes it challenging to control the amount of quantum correlations. We propose and demonstrate a method for active control of the classical-quantum crossover in magnetic insulators by applying external pressure. As a concrete example, we perform high-field, high-pressure measurements on CsCuCl₃, which has the structure of weakly-coupled spin chains. The magnetization process experiences a continuous evolution from the semi-classical realm to the highly-quantum regime with increasing pressure. Based on the idea of "squashing" the spin chains onto a plane, we characterize the change in the quantum correlations by the change in the value of the local spin quantum number of an effective two-dimensional model. This opens a way to access the tunable classical-quantum crossover of two-dimensional spin systems by using alternative systems of coupled-chain compounds.

[1] Department of Physics, Nihon University, Tokyo, Japan. [2] Department of Physics and Mathematics, Aoyama Gakuin University, Kanagawa, Japan. [3] Research Facility Center for Science and Technology, Kobe University, Kobe, Japan. [4] Graduate School of Science, Kobe University, Kobe, Japan. [5] Molecular Photoscience Research Center, Kobe University, Kobe, Japan. [6] Department of Physics, Tokyo Institute of Technology, Meguro-ku, Tokyo, Japan. [7] Institute for Solid State Physics, The University of Tokyo, Chiba, Japan. ✉email: yamamoto.daisuke21@nihon-u.ac.jp; tsakurai@kobe-u.ac.jp

Since the inception of quantum mechanics, it was recognized that the apparent dichotomy between quantum and classical physics was to be resolved, in the sense that any consistent quantum theory should retrieve the predictions of the classical theory in the limit of large quantum numbers[1]. It just so happens that unique quantum phenomena, such as quantum super-position and quantum correlation, generally become unobservable when such regime is approached. This fundamental aspect carries over to the second quantum revolution, given that quantum information and quantum technologies are based on the theory of quantum decoherence, which studies nothing but the interactions of a quantum system with a system with a large number of degrees of freedom (the environment)[2]. External control of the classical-quantum crossover would be not only intriguing, but of primary theoretical and experimental interest. A certain degree of success has been obtained in this direction with photonic[3] or optomechanical systems[4]. This work aims to demonstrate a way to achieve such control in much less flexible systems, namely a class of solid-state materials.

High-pressure application is one of the few experimental tools that can drastically change the microscopic physical parameters of materials. Effects of high pressure on material characteristics have recently been studied with considerable interest in the broad area of condensed-matter physics, having led to intriguing phenomena including pressure-driven room-temperature superconductivity[5], topological phases[6,7], and the softening of Higgs mode in spin-dimer magnets[8,9]. In particular, frustrated quantum many-body systems are promising examples expected to feel significant pressure effects since the frustration due to competing interactions gives rise to a large number of low-energy states with small energy differences, which enhance the relative impact of external pressure[10–12]. Besides, even small quantum fluctuations could also play an essential role in determining the physical properties[13–15]. Therefore, operating with external pressure on frustrated quantum materials could pave the way to actively control the amount of quantum correlations across the classical and quantum-mechanical regimes and explore exotic phenomena taking place in the crossover.

One exciting yet challenging example of frustrated quantum systems is the class of triangular-lattice antiferromagnets (TLAFs)[16]. The lattice geometry based on triangle units prohibits the standard antiferromagnetic order with an antiparallel alignment of neighboring spins. Owing to the geometrical frustration combined with magnetic anisotropy, external magnetic fields, fluctuations effects, etc., TLAF compounds exhibit a rich variety of magnetic phases[13–23]. A schematic ground-state phase diagram of two-dimentional (2D) TLAFs with exchange (or single-ion) anisotropy of easy-plane type under the magnetic field $H$ applied perpendicular to the easy plane is shown in Fig. 1a, which summarizes the well-established[14–19] and the recently predicted[20–23] theoretical results. The reciprocal of the spin quantum number, $1/S$, of magnetic ions in the material usually serves as a good indicator of the quantum correlation strength; specifically, $1/S = 2$ is the most quantum while $1/S \to 0$ is classical.

Whereas the ground state of TLAFs at some fixed parameter planes is being revealed, much less is known about what happens inside the three-variable phase diagram of Fig. 1a. There also remain other open problems, especially on essential differences between the classical (small $1/S$) and quantum (large $1/S$) regime. For example, it should be interesting if one can examine the continuous change in the nature of magnetic collective excitations from the semi-classical regime of "magnons" carrying spin-1 to the highly quantum regime of "spinons" carrying spin-1/2[24–31]. Note that the latter is expected to appear only with additional factors, such as a deformation of triangular lattice[27,28] and longer-range couplings[30], beyond the regular TLAF with nearest-

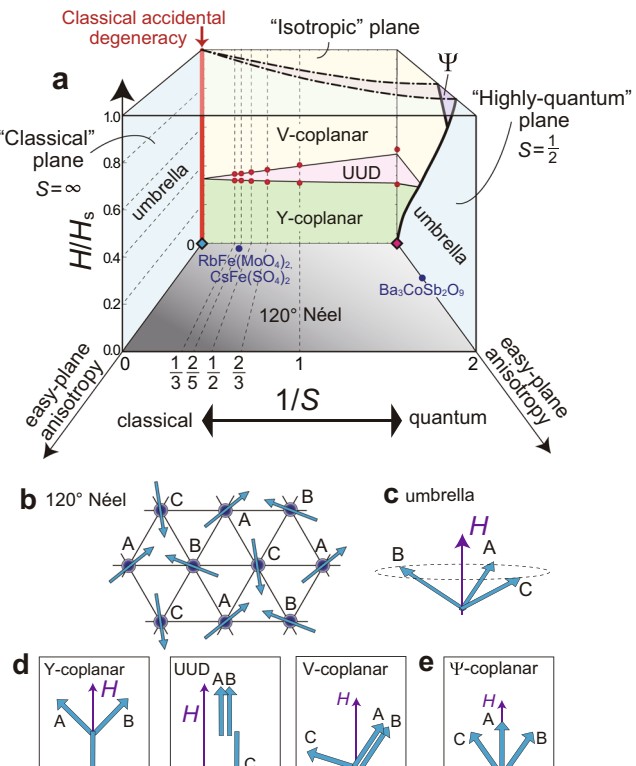

Fig. 1 Ground state of easy-plane triangular-lattice antiferromangets. a Schematic ground-state phase diagram in the space of the magnetic field $H$ scaled by the saturated field strength $H_s$, the reciprocal of spin quantum number $S$, and easy-plane anisotropy $\perp H$. The phase boundaries on the plane of no anisotropy (on the back face) are obtained by the $1/S$ expansion method with the "cutting-at-1/3" procedure[18] (solid lines) and coupled-cluster method[19] (red dots). Those on the planes of $S = 1/2$ and $H/H_s \to 1$ (on the right and top faces, respectively) are sketches based on the predictions of ref. [20] and refs. [21–23], respectively. The approximate locations of some relevant materials[69–73] are indicated by the filled circles. b-e Illustrations of the sublattice spin moments in each phase appearing in a.

neighbor interactions. Whereas "$1/S$" has been often treated as a continuous variable in the widely used analysis method, called the $1/S$ expansion[14,15,18], in real materials, however, the spin $S$ is basically fixed to a certain integer or half-integer value at the chemical synthesize. This makes it difficult to study the continuous change in the nature of materials from the classical to quantum regime.

Here we propose the concept of actively controlling the amount of quantum correlations, or more specifically, the value of "$1/S$," in a continuous manner by applying external pressure in the laboratory. The main idea is the use of materials with a coupled-chain structure, such as $ABX_3$-type hexagonal perovskites ($A$ = Rb, Cs, $B$ = V, Cr, Mn, Fe, Co, Ni, Cu, and $X$ = F, Cl, Br, I)[32–34]. Introducing a "squash" mapping, we show that the magnetic properties of coupled spin chains can be phenomenologically described by a single-layer TLAF model with effective spin $\tilde{S}$. The crucial experimental step is a series of precise magnetic measurements conducted under high pressure up to $P = 1.21$ GPa on a $CsCuCl_3$ single crystal[35–41], which allows us to determine the exchange couplings to great accuracy and, consequently, extract the parameters of the effective model. We thus demonstrate that the value of effective spin $\tilde{S}$ can be actually controlled by external pressure through the change in the material parameters. This idea of controlling the classical-quantum crossover via the squash mapping is expected to be applicable also

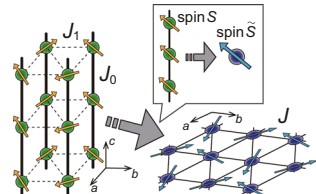

**Fig. 2 Squash mapping.** Illustration of the concept of mapping from coupled-chain model with spin $S$, intrachain coupling $J_0$, and interchain coupling $J_1$ to an effective single-layer model with spin $\tilde{S}$ and coupling $J$.

to other platforms, including cold atoms in optical lattices[42], trapped ions[43], and Rydberg atoms in arrays of optical tweezers[44], as well as directly to the other materials of the $ABX_3$-type, such as $CsNiF_3$[33] and $RbCuCl_3$[34], and to the other coupled-chain compounds with different lattice geometries.

## Results

**Coupled-chain TLAF and its squash mapping.** The hexagonal antiferromagnets of the $ABX_3$ type[32–34] have spin chains along the $c$ axis, which form triangular lattices on the $ab$ planes (see Fig. 2). We describe the magnetic properties of the coupled-chain TLAFs under magnetic fields parallel to the $c$ axis by the following Hamiltonian with spin-$S$ operators $\hat{\mathbf{S}}_{i,n}$ on site $i$ of the $n$-th triangular layer:

$$
\hat{\mathcal{H}} = -2J_0 \sum_{i,n} \left( \hat{\mathbf{S}}_{i,n} \cdot \hat{\mathbf{S}}_{i,n+1} - \Delta_0 \hat{S}_{i,n}^z \hat{S}_{i,n+1}^z \right) \\
+ 2J_1 \sum_{\langle i,j \rangle,n} \hat{\mathbf{S}}_{i,n} \cdot \hat{\mathbf{S}}_{j,n} - H \sum_{i,n} \hat{S}_{i,n}^z,
\tag{1}
$$

where the intrachain and interchain exchange couplings are assumed to be ferromagnetic and antiferromagnetic, respectively ($J_0, J_1 > 0$). Here, we took into account the possible existence of easy-plane anisotropy perpendicular to the $c$ axis ($\Delta_0 > 0$) in the intrachain coupling, which is the case for $CsCuCl_3$[35–41,45].

The key of the squash mapping is the following intuitive idea. In weakly coupled spin chains ($J_1 \ll J_0$), the time scale of the intrachain spin–spin correlations along the $c$ axis is expected to be much shorter than that of the interchain correlations in the $ab$ plane. The difference in the time scales may be characterized by the ratio of the intrachain to interchain coupling, $\alpha_J \equiv J_0/J_1$, which is ~ 5.5–6.5 for $CsCuCl_3$[36–38,40]. From the standpoint of the interchain interactions, therefore, the spins along each chain may appear to move together to make up a single "large" spin $\hat{\mathcal{S}}_i$ with an effective spin quantum number $\tilde{S} > S$, as illustrated in Fig. 2. From this intuitive idea, one could introduce the following phenomenological spin model:

$$
\hat{\tilde{\mathcal{H}}} = 2J \sum_{\langle i,j \rangle} \hat{\mathcal{S}}_i \cdot \hat{\mathcal{S}}_j + A \sum_i \left( \hat{\mathcal{S}}_i^z \right)^2 - H \sum_i \hat{\mathcal{S}}_i^z
\tag{2}
$$

with spin $\tilde{S} > S$ on a "single layer" of triangular lattice. It is natural to take into account the uniaxial two-ion exchange anisotropy along the chains by introducing uniaxial single-ion anisotropy in the effective model, given that the spins along each chain $i$ are squashed into $\hat{\mathcal{S}}_i$. The effective coupling constant $J$ and the effective anisotropy $A$ should be related to the ones in the original model as

$$
J = \frac{S}{\tilde{S}} J_1 \quad \text{and} \quad A = \frac{4S}{2\tilde{S}-1} J_0 \Delta_0,
\tag{3}
$$

such that the two models share the same value of the saturation

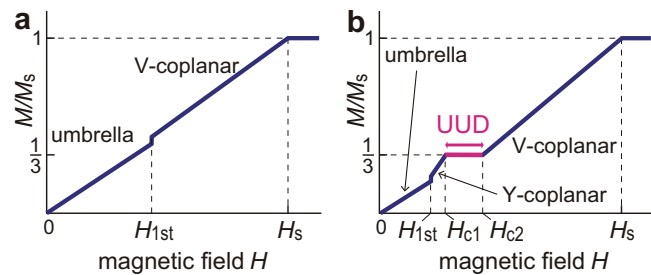

**Fig. 3 Magnetization processes.** Sketches of the magnetization possesses of the coupled-chain compound $CsCuCl_3$ **a** in the semiclassical regime under ambient or low pressure and **b** in the highly quantum regime under high pressure. The phase transition points ($H_{1st}$, $H_{c1}$, $H_{c2}$) and the saturation field ($H_s$) are marked on the horizontal axis.

magnetic field:

$$
H_s = (18J_1 + 4J_0\Delta_0)S = 18J\tilde{S} + A(2\tilde{S} - 1).
\tag{4}
$$

The fitting method for the remaining parameter $\tilde{S}$ will be discussed later for a specific case.

The above squash mapping constitutes effective dimensional reduction and spin transmutation for coupled-chain models. The effective spin quantum number $\tilde{S}$ will serve as a more suitable indicator of quantum correlation strength in weakly coupled spin chains, rather than the bare value of $S$.

**Pressure dependence of magnetic couplings in $CsCuCl_3$.** Hereafter, we take the $S = 1/2$ coupled-chain TLAF compound $CsCuCl_3$ as a specific example to pursue the subject. In $CsCuCl_3$, the intrachain coupling possesses extra Dzyaloshinskii–Moriya (DM) interaction, which causes a long-wavelength helical spin structure along the $c$ axis[46]. However, one can eliminate the DM interaction by performing a proper twist of the local spin coordinates[15] (see Supplementary Note 1 for details). When viewed in the twisted spin space, the intrachain helical spin structure appears as uniform (ferromagnetic) spin alignment along the $c$ axis, allowing us to use the model Hamiltonian in the form of Eq. (1) and to apply the squash-mapping picture shown in Fig. 2. This transformation is effectively applicable for the magnetic field $H \parallel c$, since the form of the Zeeman term is not affected by the twist along the $c$ axis.

It is well known[14] that the magnetization curve of TLAFs with strong quantum correlations exhibits a plateau structure at one-third of the saturation magnetization $M_s$ in a certain field range, $H_{c1} < H < H_{c2}$. The previous high-field experiments for $CsCuCl_3$ had reported only the existence of a first-order phase transition with no plateau for $H \parallel c$ at low temperatures[35,39,41,45], which has been interpreted as the transition from the "umbrella" to "V-coplanar" state[15,45] (Fig. 3a). The transition point $H_{1st}$ is shifted towards lower fields as the temperature increases; specifically, $H_{1st} = 12.5$ T at 1.5 K and $H_{1st} = 6$ T at 10 K[10,45]. Recently, it has been reported that applying high hydrostatic pressure $P \gtrsim 0.7$ GPa has induced the appearance of the one-third magnetization plateau[10], which has suggested the stabilization of the collinear "UUD" state and possibly the "Y-coplanar" state (Fig. 3b). The sublattice spin moments in each state are illustrated in Fig. 1c, d. The plateau formation indicates that the quantum correlations in $CsCuCl_3$ are drastically enhanced by external pressure. However, the specific pressure dependence of the Hamiltonian parameters and the microscopic origin of the plateau formation have not been revealed yet.

To quantify the pressure effects, we first perform magnetic measurements on a single crystal of $CsCuCl_3$ under hydrostatic pressure conditions up to $P = 1.21$ GPa for the temperature

dependence (below 100 K) of the magnetic susceptibility at magnetic field 1 T and the low-temperature (1.8 K) magnetization curve up to 5 T. Using the measured data shown in Fig. 4 as well as the previously reported values of the first-order transition points $H_{1st}$ at the lowest temperature (1.5 K) available in ref. [10], we quantitatively estimate the pressure dependence of the magnetic coupling parameters $J_0$, $\Delta_0$, and $J_1$ in the original model, Eq. (1), through the fittings with theoretical predictions for the ground state. For the fittings, we employ the tenth-order high-temperature expansion[47] for the magnetic susceptibility and the $1/S$-expansion method[15] for the magnetization curve and the first-order transition points. In the latter, the energy is expressed in power series of $1/S$ and anisotropy $\Delta_0$ as

$$E = S^2 E_0 + S^2 E_{\Delta_0} + S E_{LSW} + \cdots , \qquad (5)$$

where $S^2 E_0$ is the classical energy for the isotropic system. Here, we take into account up to the leading order corrections from the anisotropy, $S^2 E_{\Delta_0}$, and quantum effects within the linear spin-wave theory, $S E_{LSW}$[14,15]. The magnetization curve is obtained by $M(H) = -dE(H)/dH$[18]. The theoretical values of the magnetic field $H$ and the magnetization $M$ are converted into T (tesla) and $\mu_B/Cu^{2+}$, respectively, using the $g$ factor, which has been estimated to be 2.11 by the ESR measurements at room temperature, almost independently of pressure within the experimental precision[48]. The saturation magnetization per spin is thus given as $M_s = g\mu_B S = 1.055\mu_B$. See Methods for more details.

Figure 5a, b shows the values of $J_0$, $\Delta_0$, and $J_1$, giving the best fits between experiment and theory. Applying the least squares fittings to the values obtained at each pressure, we determine the following model functions $J_0(P)$, $\Delta_0(P)$, and $J_1(P)$ for pressure $P$ in GPa:

$$J_0(P)/k_B = 28.45 - 10.49P \text{ [K]}, \qquad (6)$$

$$\Delta_0(P) = 0.014 + 0.005P + 0.005P^2, \qquad (7)$$

$$J_1(P)/k_B = 4.86 + 2.03P \text{ [K]}. \qquad (8)$$

The values of the model functions at $P = 0$, $J_0/k_B = 28.45$ K, $\Delta_0 = 0.014$, and $J_1/k_B = 4.86$ K, are consistent with the previous estimates at ambient pressure[36–38,40]. In Fig. 5c, d, we plot the intrachain-to-interchain coupling ratio $\alpha_J(P) = J_0(P)/J_1(P)$ and the rescaled anisotropy parameter $\Delta(P) = \alpha_J(P)\Delta_0(P)$,[15,49] which characterize well the change of the material property. The parameter $\alpha_J(P)$ is strongly reduced (by half at $P \sim 1$ GPa), which indicates that a CsCuCl$_3$ crystal with weakly coupled quasi-1D spin chains turns into a more 3D system by applying hydrostatic pressure. On the contrary, the rescaled anisotropy $\Delta(P)$ experiences only a 20 percent reduction.

**Phase diagram and magnetization curve.** Using the model parameters of Eqs. ((6)–(8)) and evaluating the energies of different phases up to the leading order corrections from anisotropy and quantum effects [Eq. (5)], we obtain the theoretical ground-state phase diagram in the plane of magnetic field $H$ and pressure $P$ as shown in Fig. 6. The previous experimental observations by Sera et al.[10] on the anomalies in the magnetization curves are plotted together. Note that in the experimental data, the values of the pressure $P$ are reevaluated using the calibration scheme that we use in the current work (see Methods). The plateau endpoints for $P = 0.83$ and $0.9$ GPa are unclear within the experimental precision in ref. [10] or out of the experimental field window $H < 15$ T.

From the comparison between experiment and theory, the positions of the observed anomalies are well identified as the transition points from Y to UUD ($H_{c1}$), UUD to V ($H_{c2}$), and umbrella to the other phases ($H_{1st}$), respectively. In particular, although a narrow field range where the magnetization curve shows an almost linear increase between the first-order jump and the 1/3-plateau has not been fully identified as the Y-coplanar state only from the experiments of ref. [10], the agreement with the theoretical prediction strongly supports its existence. On the upper axis of Fig. 6, we mark the corresponding values of the effective spin $S$ in the 2D squashed model (2) (which will be addressed in the Discussion).

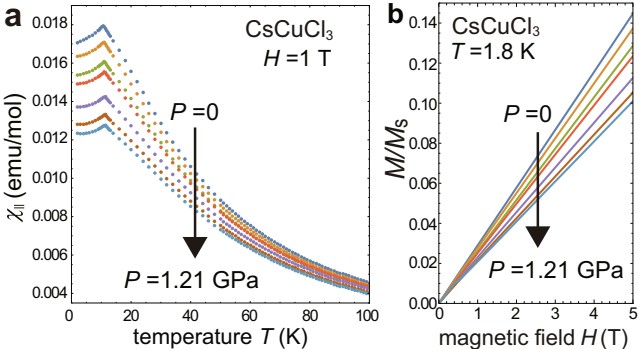

**Fig. 4 Magnetic susceptibility and magnetization curves under pressure.** **a** Longitudinal susceptibilities $\chi_{\parallel}$ at $H = 1$ T and **b** magnetization curves at $T = 1.8$ K for a CsCuCl$_3$ crystal under different pressures, $P = 0$, 0.14, 0.34, 0.49, 0.82, 1.05, 1.21 GPa (from top to bottom) when a magnetic field is applied along the $c$ axis. The magnetization $M$ is scaled by the saturation value $M_s$.

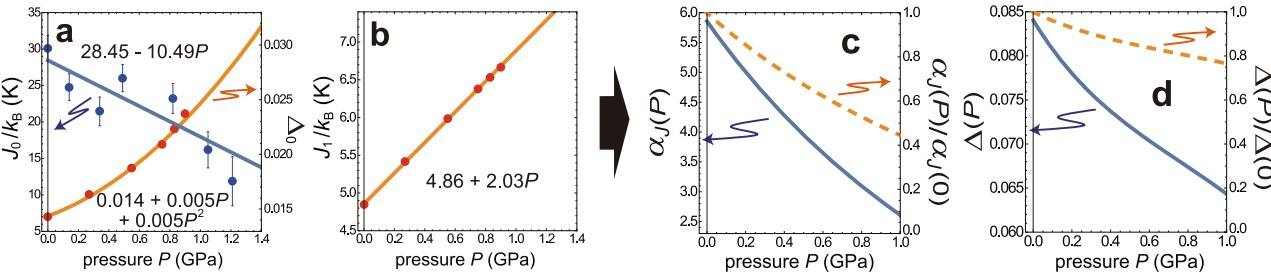

**Fig. 5 Pressure dependence of the coupling parameters in CsCuCl$_3$. a** The estimated values of the intrachain coupling constant $J_0$ and the anisotropy parameter $\Delta_0$. **b** The estimated values of the intrerchain coupling constant $J_1$. The error bars reflect six standard deviations for $J_0$ and are smaller than the symbol size in the min-max values for $\Delta_0$ and $J_1$. The model functions for each quantity, $J_0(P)$, $\Delta_0(P)$, and $J_1(P)$, are shown by the solid curves. **c, d** The ratio of the intrachain to interchain coupling, $\alpha_J(P) = J_0(P)/J_1(P)$, and the rescaled anisotropy parameter $\Delta(P) = \alpha_J(P)\Delta_0(P)$. The reduction rate from the value at $P = 0$ is plotted for each with the dashed curves.

We also compare the theoretical and experimental magnetization curves at $P = 0$, 0.75, 0.83, and 0.9 GPa in Fig. 7a–d. It can be seen that the pressure-induced change in the magnetization processes are well reproduced by the model calculations with Eqs. ((6)–(8)). While the agreement is excellent for $P \lesssim 0.75$ GPa (and still good for $P = 0.83$ GPa), it seems to get slightly worse for larger values of pressure. Especially, looking at Fig. 7d, we see that the plateau width is somewhat wider and the slope of the low-field magnetization curve is smaller than the theoretical prediction for the estimated pressure value. This might indicate that the pressure values of the experiments were slightly underestimated due to pressure inhomogeneity in the sample (see Supplementary Note 2 for a more detailed discussion).

**Mechanism for the plateau formation by applied pressure.** The width of the magnetization plateau associated with the UUD phase can be expressed as

$$W_\mathrm{p} = H_{\mathrm{c}2} - H_{\mathrm{c}1} = W_\mathrm{p}^{(\mathrm{cl})} + W_\mathrm{p}^{(\mathrm{qu})} \tag{9}$$

with

$$W_\mathrm{p}^{(\mathrm{cl})} = -16J_1 S\Delta, \tag{10}$$

$$W_\mathrm{p}^{(\mathrm{qu})} = 12J_1\big(\eta(\alpha_J) - \xi(\alpha_J)\big), \tag{11}$$

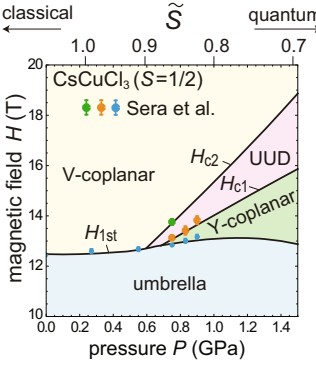

**Fig. 6 H–P phase diagram.** Theoretical ground-state phase diagram of the model for CsCuCl$_3$ in the plane of magnetic field $H$ and external pressure $P$. We mark the points at which the magnetization anomalies have been observed in the experiments of ref. [10] at temperature $T = 1.5$ K by the filled circles with error bars (blue: magnetization jump; orange: kink in between the jump and plateau; green: end point of the plateau). The corresponding values of the effective spin $\tilde{S}$ in the squashed model are indicated on the upper axis.

following the method used in a seminal work by Chubukov and Golosov[14]. Here, $\eta \equiv -\langle \hat{a}_{i_\uparrow} \hat{a}_{j_\downarrow} \rangle$ (resp. $\xi \equiv \langle \hat{a}^\dagger_{i_\uparrow} \hat{a}_{j_\uparrow} \rangle$) indicates the anomalous (resp. normal) quantum correlations between the magnons $\hat{a}_i$ on the neighboring "up" and "down" sites (resp. on the two neighboring "up" sites) in a unit triangle (see Fig. 8a). Since the relation $\eta - \xi > 0$ always holds, the quantum term $W_\mathrm{p}^{(\mathrm{qu})} > 0$ contributes to the emergence of the magnetization plateau whereas the classical term $W_\mathrm{p}^{(\mathrm{cl})} < 0$ works in the opposite way, reflecting the easy-plane anisotropy in the classical interactions between spins. The separation between the two lines, $W_\mathrm{p}^{(\mathrm{qu})} - (-W_\mathrm{p}^{(\mathrm{cl})})$, in Fig. 8b, indicates the estimation of the potential plateau width. The pressure dependence of $W_\mathrm{p}^{(\mathrm{qu})}$ and $W_\mathrm{p}^{(\mathrm{cl})}$ shows that the emergence of the plateau in CsCuCl$_3$ by applying pressure is predominantly attributed to the enhancement of quantum correlations rather than the reduction of anisotropy, reflecting the behaviors of $\alpha_J$ and $\Delta$ shown in Fig. 5c, d.

The above result shows an essential difference from the previous study[49] in the understanding of the mechanism underlying the pressure-induced plateau formation. In the analysis of ref. [49], the intrachain coupling $J_0$ was assumed to be constant with the applied pressure, and the plateau formation was explained as resulting from the reduction of the effective anisotropy $\Delta$. Our present analysis based on the parameter fittings with the experimental data has revealed that the change in $\Delta$ is not enough to explain the emergence of the plateau, but the enhancement of the quantum effects associated with the strong reduction of $J_0$ plays a key role as mentioned above. This finding leads us to the concept of the pressure-induced classical-quantum crossover, which we will discuss in the Discussion section.

## Discussion

We have studied the pressure effects on the magnetization process of the 3D material CsCuCl$_3$ with weakly coupled spin chain structure. Let us connect the results to the physics of the 2D TLAF model, Eq. (2), via the squash mapping illustrated in Fig. 2. The energy of the squashed 2D model is also expanded in power series of $1/\tilde{S}$ and anisotropy $A$ as

$$\tilde{E} = \tilde{S}^2 \tilde{E}_0 + \tilde{S}^2 \tilde{E}_A + \tilde{S}\tilde{E}_{\mathrm{LSW}} + \cdots , \tag{12}$$

in a similar fashion to Eq. (5). Substituting the correspondence relations (3), one can easily see that the classical part of the energy (scaled by the spin length) is identical for the original and effective models apart from a constant shift, that is, $S(E_0 + E_{\Delta_0}) = \tilde{S}(\tilde{E}_0 + \tilde{E}_A) + \mathrm{const.}$ for any $\tilde{S}$. Therefore, the effective spin $\tilde{S}$ should be determined in such a way that it reflects the

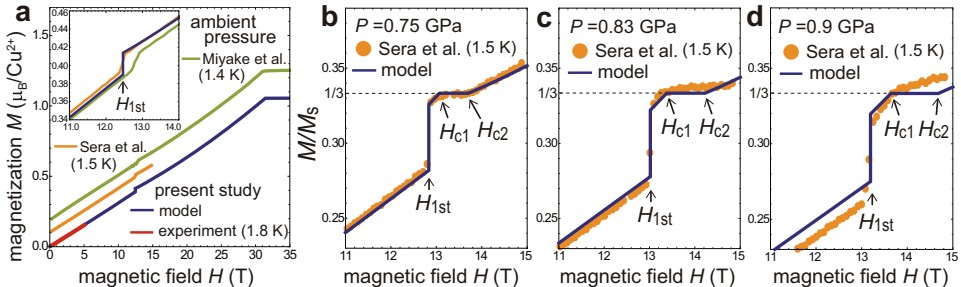

**Fig. 7 Magnetization curves.** Theoretical magnetization curves obtained by the $1/S$ expansion method[15] with the "cutting-at-1/3" procedure[18] using the model parameters $J_0(P)$, $\Delta_0(P)$, and $J_1(P)$ at temperature $T = 0$ for $P = 0$, 0.75, 0.83, 0.9 GPa, together with the corresponding experimental data of the present measurements at $T = 1.8$ K, Miyake et al. at $T = 1.4$ K (for increasing fields)[41], and Sera et al. at $T = 1.5$ K[10]. In **a** the curves are vertically shifted by 0.1 from one another to avoid overlapping. The inset shows the enlarged view of the transition region with no vertical shift. In **b–d**, the magnetization scaled by the saturation value $M_\mathrm{s}$ is plotted. Correspondingly, the curves of Sera et al. are scaled such that the plateaux are located at $M = M_\mathrm{s}/3$.

strength of quantum correlation effects. The stabilization of Y/UUD/V orders against the classical umbrella order is the most significant role of quantum correlations in TLAFs[14]. Therefore, it should be reasonable to find the value of $\tilde{S}$ such that the energy difference between the umbrella and Y/UUD/V states, $\delta E_{LSW} = E_{LSW}^{umbrella} - E_{LSW}^{Y/UUD/V}$, is well reproduced by the corresponding quantity $\delta\tilde{E}_{LSW}$ of the effective model (2). This is done by minimizing the quantity

$$\int_0^{H_s} \left| \delta E_{LSW}(H) - \delta\tilde{E}_{LSW}(H) \right|^2 dH. \tag{13}$$

A similar procedure has been used to mimic quantum fluctuation effects in 2D TLAF models by a classical-spin biquadratic coupling[50].

Before showing the result, let us comment on the difference of the squash-mapping procedure from the Weiss-field treatment in which the interactions of the spin on a given layer (ab plane) with its neighbors on adjacent layers are replaced by effective magnetic fields. Whereas such a treatment may give a reasonable

description for quasi-2D materials with small interlayer coupling[51], it fails to capture the quantum correlations in the intrachain couplings of the coupled-chain materials. The squash mapping takes into account the quantum correlations through the value of $\tilde{S}$, and more importantly, the 2D squashed model (2) is written in the same form as the model for a realistic 2D TLAF material, while the Weiss-field model includes extra terms of effective local magnetic fields with the strength and direction determined in a self-consistent fashion.

The fitting of $\delta E_{LSW}$ and $\delta\tilde{E}_{LSW}$ in the same scale of $J_1$ with respect to $\tilde{S}/S$ only depends on the intrachain/interchain coupling ratio $\alpha_J$ [under the correspondences (3)]. As expected, the value of $\tilde{S}/S$ is larger (more classical) for larger $\alpha_J = J_0/J_1$ as shown in Fig. 9a. Figure 9b are typical examples of the comparison between $\delta E_{LSW}$ and $\delta\tilde{E}_{LSW}$ with the optimized $\tilde{S}/S$ at several values of $\alpha_J$, showing a good agreement between the original (3D) and effective (2D) models. Of course, the spin operator $\hat{S}_i$ in Eq. (2) is properly defined only when $\tilde{S}$ is an integer or half-integer value in a strict sense beyond the $1/\tilde{S}$ expansion. Nevertheless, the value of $\tilde{S}$ can still be taken as an indicator for the strength of quantum fluctuations existing in the coupled-chain compound under consideration. For example, a material with the intrachain coupling $J_0$ being five times larger than the interchain coupling $J_1$ is expected to exhibit the same extent of quantum effects as the corresponding 2D material with the spin being about two times larger than the original one.

Using the result of Fig. 9a with the original spin value $S = 1/2$, we can translate the pressure dependence of the intrachain/interchain coupling ratio $\alpha_J$ for CsCuCl$_3$, shown in Fig. 5c, into continuous change of the effective spin $\tilde{S}$ in terms of the 2D TLAF model. The obtained values of $\tilde{S}$ are indicated on the upper axis of the phase diagram in Fig. 6. Now let us discuss the extension of the model calculations beyond the parameter range of the current experiments, with the caveat that the extrapolation is in general less reliable. Figure 10 shows the predicted phase diagram in an extended parameter space, where the horizontal axis is converted from $P$ to $1/\tilde{S}$. The corresponding values of $\alpha_J$ are indicated on the upper axis. When $\alpha_J = 0$, the model is trivially reduced to the spin-1/2 Heisenberg model for a purely 2D TLAF with isotropic

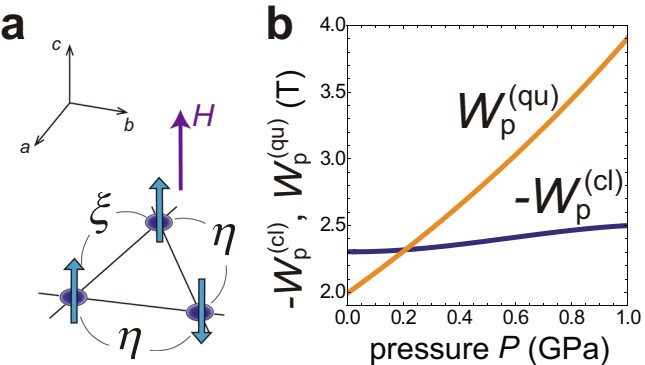

**Fig. 8 Two contributions to the plateau width. a** Quantum fluctuation measures $\eta$ and $\xi$ in the UUD state. **b** Classical and quantum contributions, $W_p^{(cl)}$ and $W_p^{(qu)}$, to the plateau width as functions of pressure $P$. We plot $W_p^{(cl)}$ with the negative sign for convenience.

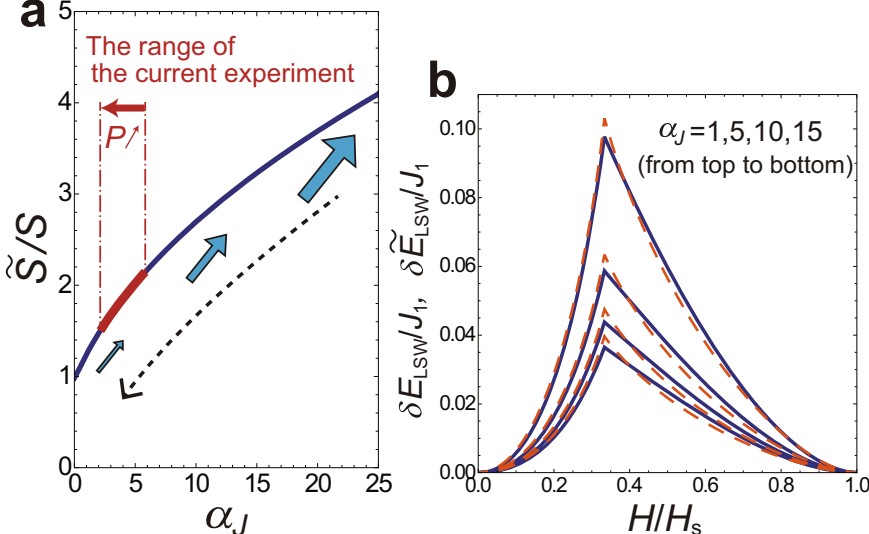

**Fig. 9 Quantum correlation measure for coupled spin chains. a** Increase rate of the effective spin $\tilde{S}$ in the squashed model from the original value $S$ as a function of the ratio of the intrachain to interchain coupling, $\alpha_J = J_0/J_1$. **b** Energy differences between the umbrella and Y/UUD/V states, $\delta E_{LSW}$ for the original model with $\alpha_J = 1, 5, 10, 15$ (blue-solid lines) and $\delta\tilde{E}_{LSW}$ for the squashed model with $\tilde{S}/S = 1.24, 2.02, 2.69, 3.23$ (orange-dashed lines), within the leading $1/S$ (linear spin-wave) corrections.

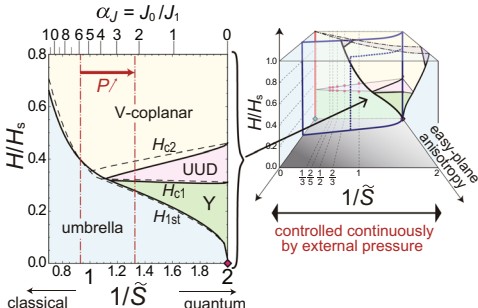

**Fig. 10 Phase diagram in an extended parameter space.** Theoretical ground-state phase diagram of the model for $CsCuCl_3$ in the plane of magnetic field $H$ scaled by the saturation value $H_s$ and the reciprocal of the effective spin $\tilde{S}$. The solid and dashed curves in the left panel are the phase boundaries obtained for the original model of $CsCuCl_3$ with the ratio of the intralayer to interlayer interaction, $\alpha_J$ and the effective 2D model with the corresponding values of the effective spin $\tilde{S}$, respectively. The range of the pressure application in the current experiments is indicated by the vertical red dashed-dotted lines. The right illustration schematically shows the corresponding parameter plane in the three-variable phase diagram for 2D easy-plane triangular-lattice antiferromangets, shown in Fig. 1.

exchange coupling. Therefore, the pressure-induced stabilization of the magnetization plateau can be interpreted by means of the effective 2D TLAF model as a consequence that the pressure pushes the value of $1/\tilde{S}$ from the semi-classical ($1/\tilde{S} < 1$) regime towards the highly quantum ($1/\tilde{S} = 2$) regime. Although the change in $1/\tilde{S}$ was not significantly large in the current experiment with a piston cylinder cell, it was fortunate that the magnetic parameters of $CsCuCl_3$ at ambient pressure were located in the vicinity of the crossover regime between the semi-classical and highly quantum magnetization processes, which are shown in Fig. 3a, b, respectively.

Note that, as shown in Fig. 10, whereas the effective 2D model reproduces well the phase boundaries $H_{c1}$ and $H_{1st}$ in low fields, the value of $H_{c2}$ is somewhat overestimated. This is caused by the fact that the fitting of the zero-point energies, $\delta E_{LSW}$ and $\delta \tilde{E}_{LSW}$, is relatively less satisfactory in the high-field region, as seen in Fig. 9b, which could be improved by considering the $H$ dependence of the effective spin $\tilde{S}$ but with extra complexity.

To conclude, through high-pressure magnetic measurements and theoretical investigations on a $CsCuCl_3$ crystal, we have developed a scientific concept for the control of quantum-mechanical correlations in weakly-coupled spin chain materials by applying external pressure. The parameter fitting for the model Hamiltonian of $CsCuCl_3$ has shown that the ratio of the intra-chain to interchain spin coupling, $\alpha_J$, is strongly reduced by hydrostatic pressure application. From an intuitive idea of mapping the spins along each chain into a single large spin $\tilde{S} > S$, we introduce an effective spin model that is "squashed" onto a 2D plane and establish the correspondence between the parameters of the original and effective model Hamiltonians. Since the spin quantum number can take only an integer or half-integer value in nature, one can in principle access the phase diagram only with discrete values of $S$ in experiments. Our observations open up an interesting possibility of performing quantum simulation studies that can interpolate the properties of 2D spin models at discrete spin values by performing high-pressure experiments on coupled-chain compounds. Moreover, the spin value $S$ has been actually treated as a continuous variable in theoretical studies using analytical methods such as the $1/S$ expansion and the Schwinger-boson mean-field theory (with parameter $\kappa = 2S$[13,52–55]). The interpretation based on the squash mapping opens a way for

high-pressure experiments on coupled-chain compounds to directly realize a huge variety of the theoretical phase diagrams that has been predicted so far (and will be obtained in the future) for 2D models with continuous $S$.

Considering the variety of coupled-spin-chain compounds, including the other materials in the $ABX_3$-type hexagonal perovskite family[32–34] and those with different lattice geometries, this concept also provides us with a unique opportunity to study the continuous classical-to-quantum crossover of the ground state and the elementary excitations in a wide variety of 2D frustrated quantum antiferromagnets. For example, the spatially anisotropic TLAF model has been extensively studied in the literature[28,56–62] as a model showing a rich phase diagram including quantum spin liquids. High-pressure experiments on a coupled-chain compound, e.g., $RbCuCl_3$, in which spin-1/2 chains form a spatially anisotropic triangular lattice[34], could enable us to simulate the theoretical phase diagram with active and continuous control of effective spin $\tilde{S}$. Such an experiment may allow access to the spin liquid quantum critical point via the melting of magnetic long-range order by tuning pressure (or the value of $\tilde{S}$). Future research in such a direction would be promising to shed new light into the connection between the semi-classical "magnon" and highly quantum "spinon" descriptions of magnetic quasiparticles[24–31]. Finally, note that although not a few compounds in the family of $ABX_3$-type hexagonal perovskites have antiferromagnetic intra-chain coupling[32], there is still every chance that the pressure application changes it to ferromagnetic one, allowing for the squashed model description we proposed here.

## Methods

**Sample setting and magnetization measurements under pressure**. Single crystal samples of $CsCuCl_3$ were prepared by following the procedure described in ref. [63]. A clamp-type piston–cylinder pressure cell made of CuBe alloy with an outer diameter of $8.7\phi$, an inner diameter of $2.7\phi$ and a cylinder length of 72 mm was used[64]. A sample is enclosed in a Teflon capsule with a pressure medium Daphne 7373 (Idemitsu Kosan Co., Ltd.). A plate-like $CsCuCl_3$ sample with the long axis along the $c$-axis was prepared. The dimension was 2 mm × 6 mm and the thickness was about 1 mm (~18 mg). The pressure was calibrated by the change of the superconducting transition temperature of tin[65]. A tin foil with a thickness of 0.2 mm was formed into a tube shape (~30 mg), and the sample was placed in this tube.

Magnetization was measured by a commercially available magnetometer equipped with a superconducting quantum interference device (MPMS-XL, Quantum Design, Inc.). The measurement was performed using the option "background subtraction" of MultiVu software attached to MPMS. First, to obtain the background data, temperature variation and magnetic field variation sequences were run at ambient pressure for the pressure cell including tin without sample. Then, the magnetization of $CsCuCl_3$ at each pressure was obtained by subtracting the background from the total magnetization including $CsCuCl_3$ sample in the same sequences. The background data at ambient pressure was used for all measurements. The magnetic field is applied parallel to the $c$-axis. The temperature variation measurements were done at 1 T below 100 K, and the field variation measurements were done at 1.8 K up to 5 T.

In temperature variation measurement, the temperature range was limited below 100 K to avoid change in pressure. The clamp-type pressure cell has a relatively large pressure drop when the temperature is decreased, especially between the room temperature and 100 K (at most 0.2 GPa), whereas it hardly has change in pressure below 100 K[66].

In this study, the pressure was calibrated using the relationship between the pressure and the superconducting transition temperature of tin given in ref. [65]. In the magnetization measurement under pressure by Sera et al.[10], the pressure was also calibrated by the superconducting transition temperature of tin, but by a different formula given in ref. [67]. The pressure values stated when we referred to the data of Sera et al.[10], including Figs. 6 and 7, were the ones recalibrated by the former calibration formula; specifically, $P = 0.25$, 0.50, 0.68, 0.75, and 0.81 GPa in Sera et al. were reevaluated as $P = 0.27$, 0.55, 0.75, 0.83, and 0.90 GPa, respectively, and $P = 0.1$ MPa was regarded as $P = 0$.

**Parameter fitting of magnetic susceptibility data**. Figure 4a shows the temperature dependence of the magnetic susceptibility parallel to the $c$ axis, $\chi_{\parallel}$, measured at $H = 1$ T under different pressures, $P = 0$, 0.14, 0.34, 0.49, 0.82, 1.05, and 1.21 GPa. The core diamagnetic ($\chi_{dia} = -1.09 \times 10^{-4}$ emu/mole) and Van-Vleck paramagnetic ($\chi_v = 0.48 \times 10^{-4}$ emu/mole) contributions[36] are already subtracted.

As can be seen, the overall value of $\chi_\parallel$ decreases considerably as the pressure increases, which indicates that the dominant coupling parameter for the magnetic energy scale, namely the intrachain coupling strength $J_0$, significantly decreases. The peak of each curve is located at the Néel temperature $T_N$.

To quantify the pressure dependence of $J_0$, we perform a fitting of the experimentally measured $\chi_\parallel(T)$ in the temperature range 50–100 K to the expression

$$\frac{T}{C}\chi_\parallel(T) = \text{Padé}(4,5)\left[1 + \sum_{n=1}^{9} a_n\left(\frac{J_0}{k_B T}\right)^n\right], \quad (14)$$

where, $C = N_0 g^2 \mu_B^2 / 4k_B$ is the Curie constant with $N_0$ being the Avogadro number and Padé(4,5)[···] means the Padé approximant of order [4/5]. The coefficients $a_n$, which are (lengthy) functions of $\alpha_J$, are obtained by the tenth-order high-temperature expansion method[47] (see Supplementary Fig. 1). Here, we ignored the small contributions from $\Delta_0$. In Fig. 5a, the values of $J_0$ obtained by the fittings were shown. Note that the values of $J_1$ fitted to the magnetic susceptibility data strongly vary depending on the temperature range used for the fittings. Therefore, we adopt only the model function of $J_0(P)$, which is the most dominant parameter for the susceptibility measurements, from the above fittings.

**Parameter fitting of magnetization curves**. Figure 4b shows the scaled magnetization curves $M/M_s$, which are measured under static magnetic field up to 5 T at temperature $T = 1.8$ K for different pressures, $P = 0, 0.14, 0.34, 0.49, 0.82, 1.05,$ and 1.21 GPa. It can be seen that the curves are almost linearly proportional to $H$ in this field range. The magnetization curves have also been measured by Miyake et al.[41] (at $P = 0$) and Sera et al.[10] (up to $P = 0.90$ GPa in our calibration). There is a little variability in the slope of $M(H)$ among the experiments.

The model functions of $\Delta_0$ and $J_1$ [Eqs. (7) and (8)] were determined such that the low-temperature magnetization curves obtained by the different experiments could be all reasonably reproduced (see Fig. 7 and Supplementary Fig. 2). The theoretical calculations were based on the evaluation of the energy up to the leading orders of the anisotropy and $1/S$. Each term in Eqs. (5) and (12) is obtained by following the procedure of ref. 15 for each phase (umbrella, Y, or V). It should be noted that the magnetic field $H$ and single-ion-type anisotropy $A$ have to be treated as order of $S$ and $S/(2S-1)$[68], respectively, to obtain the correct expression for the saturation field. The slope of the magnetization curve in low fields can be calculated from the thermodynamic relation $M = -dE/dH$[18] with $E$ for the umbrella phase.

## Data availability
The data that support the findings of this study are available from the corresponding author upon reasonable request.

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

## Acknowledgements
We would like to thank Prof. Masafumi Sera for giving us the details of their pressure calibration method. We also thank Dr. Giacomo Marmorini for a careful reading of the manuscript. This work was carried out by the joint research program of Molecular Photoscience Research Center, Kobe University, and supported by KAKENHI from Japan Society for the Promotion of Science: Grant Numbers 18K03525 (D.Y.), 19K03746 (T.S.), 19K21852 (H.O.), 17H01142 (H.T.), and 19H00648 (Y.U.), and "Early Eagle" grant program from Aoyama Gakuin University Research Institute (D.Y.).

## Author contributions
D.Y., T.S., S.O., and H.O. designed and coordinated the studies. T.S. and R.O. performed the experiments. T.S. and Y.U. developed the pressure cell used in the experiments. H.T. grew CsCuCl₃ single crystals. D.Y. built the theoretical model to analyze the experimental results and wrote the main text with input from the other authors. The Methods section was written by T.S. All authors read and approved the paper.

## Competing interests
The authors declare no competing interests.
