## [Peer Review File · Nature Communications]

Reviewers' Comments:

Reviewer #1:

Remarks to the Author:

The manuscript under consideration represents an interesting collaborative study of magnetization physics of quantum magnet material subject to high pressure. The agreement between experimental data and theoretical fits based on the proposed model Eq.2, as shown in Fig.7, is quite impressive. Both experimental and theoretical data are unusual and should be of interest to many investigators of quantum magnetism.

I, therefore, recommend the manuscript for publication, even though I do want to ask a couple of questions.

The first one is experimental - the authors mention that CsCuCl₃ possesses Dzyaloshinskii-Moriya interaction along the c-axis, which is taken into account by the appropriate unitary transformation (as was done previously in Ref.15). This certainly works for the case of the magnetic field oriented along the c-axis. It is however not clear from the manuscript if the authors studied what happens when the field is applied perpendicular to the c-axis? Do the magnetization and its flat features disappear in this geometry?

The second question is theoretical and has to do with the concept of the effective spin \tilde{S} in Eq.2, which is found to decrease with the pressure (Fig.6 and 10). Is it possible to think of it as being due to the effective Weiss field experienced by the spin in a given layer from its neighbors in adjacent layers? Such an effective field would be proportional to J_0 and will have to be determined self-consistently. But does this way of thinking makes sense at all? I hope the authors can comment on this.

Reviewer #2:

Remarks to the Author:

The authors present a phenomenological model of triangular-lattice antiferromagnets (TLAFs) tuned between quantum and classical behaviour by the application of pressure, supported by high-pressure experimental magnetisation data of the relevant antiferromagnet CsCuCl₃. Their model consists of flattening the ferromagnetic c chains into an effective 2D single layer with a new effective spin and coupling between these new effective moments. The authors claim that this model allows for the exploration of quantum crossover in two-dimensional systems using such coupled-chain materials as a basis.

With it being the system in which the authors are demonstrating their model, I am not certain that the authors address the ambient pressure helical magnetic structure of CsCuCl₃ in sufficient detail. They acknowledge the existence of chains along the c direction as ferromagnetic but neglect that the moment direction is helically modulated over a long period (K. Adachi et al., J. Phys. Soc. Jpn. 49, 545 (1980)). Being that the present authors' model involves collapsing along these chains to a plane, neglecting this helical structure and treating the chains are simply ferromagnetic seems a notable omission, at least to justify that the used approach is still applicable. The authors do mention in their conclusion compounds with antiferromagnetic intra-chain coupling but I believe it is necessary to address the helical case as it is applicable for the material used in these experiments, and it will strengthen their conclusions speculating about other spin-chain compounds.

In CsCuCl₃ specifically, the authors acknowledge from the literature the emergence of new magnetic phases under the application of magnetic field along the c-direction, and "the existence of a first-order phase transition at $H_{1st} \approx 12.5$ T with no plateau for $H \parallel c$ ". I am unclear how this is entirely consistent with the report of Schotte et al. (U. Schotte et al., J. Phys.: Condens. Matter 6,

10105 (1994)) which appears to show a similar transition at 5.6 T for $H \parallel c$. If, as I believe from the magnetic field-temperature phase diagram of Miyake et al. in the work cited by the present authors, this is purely an effect of different measurement temperatures, I think this temperature dependence should be noted in this work. Schotte et al. further discuss the ground states of such systems under applied field as this as the present authors show in Fig. 1 (and do acknowledge with ref. 15) and seems directly relevant.

The authors' model itself is clearly presented and characterised in terms of the parameter $\alpha_{J_0} \equiv J_0/J_1$, and the effect of pressure on this parameter in CsCuCl₃ is presented. The authors claim these measurements are an effective proof of concept for their model of modifying the spin S of a system continuously rather than by the half-integer values accessible by traditional methods. The authors then claim that their model will be useful when applied to other systems with ferromagnetic chains to reveal theoretical phase diagrams.

The high-pressure magnetisation measurements performed using a SQUID piston-cylinder type cell are well explained and the necessary consideration of background subtraction to achieve accurate measurement of the sample is demonstrated, alongside a useful explanation of the manometry used. The additional recalibration of the data of Sera et al. is very useful for their comparison in this present work. The use of the ambient pressure background measurement for subtraction for all sample pressures is most likely sufficient here. From the experimental details provided I am pleased that the authors have considered and performed this experimental aspect of their report accurately.

The authors' claim that Fig. 7 shows "that the pressure-induced change in the magnetization processes are well reproduced by the model calculations with Eqs.(6-8)" does not seem immediately accurate. The presented experimental data of Sera et al. in Fig. 7(d) at 0.9 GPa does not clearly show the plateau corresponding to the stabilisation of UUD order to the extent the current model predicts (also for 0.83 GPa), whereas I agree that the 0.75 GPa data agrees quite well. I believe the authors should address this disparity, as the size of the plateau is expected to increase with pressure and so naively I would expect it to be more clearly visible in the data here, either with a comment on the previous experimental data or on the model itself. The authors do state that "The plateau endpoints for $P = 0.83$ and 0.9 GPa are unclear within the experimental precision in Ref. 10 or out of the experimental field window $H < 15$ T" and somewhat address the failings of the model later in reference to figure 10 (from line 391), but I believe that to claim the present model specifically reproduces these results well requires further clarification.

The figures included in the supplementary information serve well to illustrate the authors' points where referenced and I find overall the figures throughout the manuscript to be effective and well made.

I do believe that this work is of significant interest to the low-dimensional and quantum magnetism communities and presents a particularly interesting approach to the study of two-dimensional magnetic models using a three-dimensional system as a starting point. Additional methods by which to study the evolution of two-dimensional systems will be of use for a number of relevant questions in the field, and as the authors point out, the variety of spin-chain compounds provides ample opportunities to explore this further.

With more careful justification of the applicability of the proposed model to CsCuCl₃ given its helical magnetic structure as a priority, as well as a more complete comparison of the model predictions with previously published experimental results, I believe that this work will be of high quality and significant interest which justifies publication in Nature Communications.

Reviewer #3:

Remarks to the Author:

One of the barriers that must be overcome in quantum computing, quantum transmission, etc., is the control of quantum spin and classical spin. This is an excellent paper that contains the important information needed for this and attempts to clarify it entirely from experimental facts. Figure 1 in particular is educational and informative for many concerned researchers. Therefore, I recommend to submit.

1 Summary of changes

1. All changes in the manuscript text file (to address the referees' comments) are highlighted in red in the enclosed file "manuscriptRED.pdf".
2. Supplementary Notes 1 and 2 have been newly added in the Supplementary information file.
3. Refs. 46, 51, and 56 have been newly added.

2 Response to Reviewer #1

Reviewer: "The manuscript under consideration represents an interesting collaborative study of magnetization physics of quantum magnet material subject to high pressure. The agreement between experimental data and theoretical fits based on the proposed model Eq.2, as shown in Fig.7, is quite impressive. Both experimental and theoretical data are unusual and should be of interest to many investigators of quantum magnetism. I, therefore, recommend the manuscript for publication, even though I do want to ask a couple of questions."

Response: We thank Reviewer #1 for his/her detailed attention to our new findings and for recommending our paper for publication in Nature Communications. Please find our answers to his/her questions below.

Reviewer: "The first one is experimental - the authors mention that CsCuCl₃ possesses Dzyaloshinskii-Moriya interaction along the c -axis, which is taken into account by the appropriate unitary transformation (as was done previously in Ref.15). This certainly works for the case of the magnetic field oriented along the c -axis. It is however not clear from the manuscript if the authors studied what happens when the field is applied perpendicular to the c -axis? Do the magnetization and its flat features disappear in this geometry?"

Response: In the present work, we have focused only on the case of $H \parallel c$ (both experimentally and theoretically) with the purpose of establishing the concept of classical-quantum crossover controlled by pressure for general coupled-chain materials. The specific example, CsCuCl₃, possesses extra Dzyaloshinskii-Moriya interaction along the c -axis, but it can be eliminated effectively by the "twisted spin" unitary transformation for $H \parallel c$.

As the reviewer correctly points out, such a transformation does not work well when the field is applied perpendicular to the c -axis. As an answer to the reviewer's question, in that case, the flat feature (plateau) in the magnetization curve is not allowed to appear in a strict sense because the Hamiltonian and the transverse-field term are not commutative (the magnetization is no longer a preserved quantity). The previous work by Sera et al. [10] has reported some experimental results for the pressure dependence of the magnetization process also for $H \perp c$, and has found several different incommensurate spin phases. However, it seems that no systematic and simple physics that is directly related to the classical-quantum crossover of 2D TLAfs is found under pressure for $H \perp c$ (due to the "ineliminable" DM interaction), in contrast to our $H \parallel c$ case. To our best knowledge, no theoretical studies have been performed on the pressure dependence of CsCuCl₃ for $H \perp c$. Studying this is out of the scope of the present paper, but will be an interesting future work for the specific material.

In the first paragraph of the section “Pressure dependence of magnetic couplings in CsCuCl₃” of the revised manuscript, we made some more explanation on the existence of the Dzyaloshinskii-Moriya interaction in CsCuCl₃ and how to treat it in the model. Moreover, for reader’s convenience, we present a brief review of Nikuni et al. [15] in the newly-added “Supplementary Note 1”, regarding the unitary transformation to eliminate the DM term in the case of $H \parallel c$ (and explain why it is not applicable for $H \perp c$). Please see the part highlighted in red (lines 179-190) of the revised manuscript and the Supplementary information file.

Reviewer: “The second question is theoretical and has to do with the concept of the effective spin \tilde{S} in Eq.2, which is found to decrease with the pressure (Fig.6 and 10). Is it possible to think of it as being due to the effective Weiss field experienced by the spin in a given layer from its neighbors in adjacent layers? Such an effective field would be proportional to J_0 and will have to be determined self-consistently. But does this way of thinking makes sense at all? I hope the authors can comment on this.”

Response: We thank the reviewer for raising this interesting point. The conventional Weiss-field description and our interpretation with introducing the effective spin quantum number \tilde{S} are similar in the sense that the increase of the quantum fluctuations by pressure can be explained via the decrease of the intrachain coupling J_0 . While in the former the effective magnetic fields that stabilize the classical spin ordering get decreased by pressure, in the latter the increase of the quantum correlations is more directly taken into account as the decrease of \tilde{S} .

The difference between the two treatments is, firstly, that the Weiss approximation overestimates the classical spin orders (or underestimates the quantum fluctuations), whereas the squash mapping onto a 2D model properly takes into account the amount of quantum fluctuations by fitting the original and mapped models with respect to the value of \tilde{S} . More importantly, whereas the Weiss model includes extra terms of effective local magnetic fields applied in “unrealistic” directions (determined in a self-consistent fashion), the squashed model is written in the same form as the model for a realistic 2D TLAF material in a realistic uniform magnetic field. This is indeed essential for the main idea of our work that the classical-quantum crossover of 2D spin systems is studied using the variety of spin-chain compounds under controlled pressure.

We added a new paragraph in the Discussion to mention the above point. Please see the part highlighted in red (lines 367-382)

We believe that the detailed explanations we added in the revised manuscript improved the readability and quality of the paper, thanks to the reviewer’s questions.

3 Response to Reviewer #2

Reviewer: “The authors present a phenomenological model of triangular-lattice antiferromagnets (TLAFs) tuned between quantum and classical behaviour by the application of pressure, supported by high-pressure experimental magnetisation data of the relevant antiferromagnet CsCuCl₃. Their model consists of flattening the ferromagnetic c chains into an effective 2D single layer with a new effective spin and coupling between these new effective moments. The au-

thors claim that this model allows for the exploration of quantum crossover in two-dimensional systems using such coupled-chain materials as a basis.”

“The authors’ model itself is clearly presented and characterised in terms of the parameter $\alpha_J \equiv J_0/J_1$, and the effect of pressure on this parameter in CsCuCl₃ is presented. The authors claim these measurements are an effective proof of concept for their model of modifying the spin S of a system continuously rather than by the half-integer values accessible by traditional methods. The authors then claim that their model will be useful when applied to other systems with ferromagnetic chains to reveal theoretical phase diagrams.

The high-pressure magnetisation measurements performed using a SQUID piston-cylinder type cell are well explained and the necessary consideration of background subtraction to achieve accurate measurement of the sample is demonstrated, alongside a useful explanation of the manometry used. The additional recalibration of the data of Sera et al. is very useful for their comparison in this present work. The use of the ambient pressure background measurement for subtraction for all sample pressures is most likely sufficient here. From the experimental details provided I am pleased that the authors have considered and performed this experimental aspect of their report accurately.”

“The figures included in the supplementary information serve well to illustrate the authors’ points where referenced and I find overall the figures throughout the manuscript to be effective and well made.

I do believe that this work is of significant interest to the low-dimensional and quantum magnetism communities and presents a particularly interesting approach to the study of two-dimensional magnetic models using a three-dimensional system as a starting point. Additional methods by which to study the evolution of two-dimensional systems will be of use for a number of relevant questions in the field, and as the authors point out, the variety of spin-chain compounds provides ample opportunities to explore this further.

With more careful justification of the applicability of the proposed model to CsCuCl₃ given its helical magnetic structure as a priority, as well as a more complete comparison of the model predictions with previously published experimental results, I believe that this work will be of high quality and significant interest which justifies publication in Nature Communications.”

We thank Reviewer #2 for his/her work in carefully reading our manuscript and for recognizing that our paper will be of high quality and significant interest which justifies publication in Nature Communications, after some more justification. We hope that the answers to the Reviewer’s questions below resolve his/her remaining concerns.

Reviewer: “With it being the system in which the authors are demonstrating their model, I am not certain that the authors address the ambient pressure helical magnetic structure of CsCuCl₃ in sufficient detail. They acknowledge the existence of chains along the c direction as ferromagnetic but neglect that the moment direction is helically modulated over a long period (K. Adachi et al., J. Phys. Soc. Jpn. 49, 545 (1980)). Being that the present authors’ model involves collapsing along these chains to a plane, neglecting this helical structure and treating the chains as simply ferromagnetic seems a notable omission, at least to justify that the used approach is still applicable. The authors do mention in their conclusion compounds with anti-ferromagnetic intra-chain coupling but I believe it is necessary to address the helical case as it is applicable for the material used in these experiments, and it will strengthen their conclusions speculating about other spin-chain compounds.”

Response: As the reviewer points out, the CsCuCl_3 possesses Dzyaloshinskii-Moriya interaction along the c -axis and, as a result, the spin moment direction is helically modulated over a long period. However, this helical modulation can be effectively eliminated by performing a proper unitary transformation from the laboratory frame to a “twisted spin” frame. In the twisted spin frame, the helical magnetic structure of CsCuCl_3 appears just as ferromagnetically-ordered spins along the c axis. At the same time, under the unitary transformation, the model Hamiltonian is properly reduced to the form of Eq. (1) with no Dzyaloshinskii-Moriya term. (Note that such a treatment of the helical structure has already been established by Nikuni and Shiba in Ref. [15]). Therefore, the application of our squash mapping to the helical case of CsCuCl_3 is wholly justified.

In the section “Pressure dependence of magnetic couplings in CsCuCl_3 ” of the revised manuscript, we added a more detailed explanation on why we can use the case of CsCuCl_3 as a demonstration of our squash mapping. Please see the part highlighted in red (lines 179-190). In addition, more technical details of the unitary transformation (firstly proposed in Ref. [15]) are presented in the newly-added “Supplementary Note 1” in the Supplementary information file for reader’s convenience. We believe that those additions provide further justification for our model analysis based on the Hamiltonian Eq. (1) and the squash mapping for the ferromagnetically-ordered spins along the c axis.

Reviewer: “In CsCuCl_3 specifically, the authors acknowledge from the literature the emergence of new magnetic phases under the application of magnetic field along the c -direction, and “the existence of a first-order phase transition at $H_{1\text{st}} \approx 12.5$ T with no plateau for $H \parallel c$ ”. I am unclear how this is entirely consistent with the report of Schotte et al. (U. Schotte et al., J. Phys.: Condens. Matter 6, 10105 (1994)) which appears to show a similar transition at 5.6 T for $H \parallel c$. If, as I believe from the magnetic field-temperature phase diagram of Miyake et al. in the work cited by the present authors, this is purely an effect of different measurement temperatures, I think this temperature dependence should be noted in this work.”

Response: We thank the reviewer for the suggestion. We added a comment on the temperature dependence of the first-order phase transition point into the lines 199-202, and added the paper “U. Schotte et al., J. Phys.: Condens. Matter 6, 10105 (1994)” into the reference list as Ref. 46. The difference in the transition field strength comes from different measurement temperatures, as the referee points out. The value $H_{1\text{st}} \approx 12.5$ T was obtained at the temperature ~ 1.5 K, while 5.6 T in the report by Schotte et al. was measured at ~ 10 K. The temperature dependence of $H_{1\text{st}}$ can be clearly seen in Fig. 4(a) of “Sera et al., Phys. Rev. B **96**, 014419 (2017)” (Ref. [10] of our manuscript).

In the present work, we compared the theoretical model calculations with the lowest-temperature experimental data ever reported (~ 1.5 K) in order to study the pressure dependence of the magnetization process in the highly-quantum regime.

Reviewer: “Schotte et al. further discuss the ground states of such systems under applied field as this as the present authors show in Fig. 1 (and do acknowledge with ref. 15) and seems directly relevant.”

Response: We thank again the reviewer for pointing out the lacking of this important reference. In the revised manuscript, we properly referred to the analysis by Schotte et al. on the ground states in the lines 198-199.

Reviewer: “The authors’ claim that Fig. 7 shows “that the pressure-induced change in the

magnetization processes are well reproduced by the model calculations with Eqs.(6-8)” does not seem immediately accurate. The presented experimental data of Sera et al. in Fig. 7(d) at 0.9 GPa does not clearly show the plateau corresponding to the stabilisation of UUD order to the extent the current model predicts (also for 0.83 GPa), whereas I agree that the 0.75 GPa data agrees quite well. I believe the authors should address this disparity, as the size of the plateau is expected to increase with pressure and so naively I would expect it to be more clearly visible in the data here, either with a comment on the previous experimental data or on the model itself. The authors do state that ”The plateau endpoints for $P = 0.83$ and 0.9 GPa are unclear within the experimental precision in Ref. 10 or out of the experimental field window $H < 15$ T” and somewhat address the failings of the model later in reference to figure 10 (from line 391), but I believe that to claim the present model specifically reproduces these results well requires further clarification.”

Response: As the reviewer points out, the agreement between our theory and the experimental magnetization curve reported previously by Sera et al. in Ref. [10] seems to get slightly worse for higher pressures, especially at 0.9 GPa in Fig. 7(d). On this point, we had private communication with Dr. Masafumi Sera, one of the main authors of Ref. [10].

Through the discussion, we found the fact that the sample size used in their measurements was about three times longer than the one we used. In general, the pressure inhomogeneity in a sample becomes larger as the sample gets longer since it occurs along the cylindrical axis in the piston-cylinder type pressure cell when pressure-transmitting fluid freezes. In addition, Sera et al. measured the pressure only at the bottom of the sample while we properly measured the pressure experienced by the sample by putting the sample into a tin tube. Those two factors, namely, the relatively large pressure inhomogeneity and the measurement method for the pressure value could cause underestimation of the pressure in the experiment of Ref. [10]. If so (i.e., the true pressure value is slightly larger than the estimated value by Sera et al.), the slight discrepancy from the theoretical curve in Fig. 7d could naturally be explained since the plateau width is larger for higher pressures. Further careful future experiments under high magnetic fields with precise estimation of the pressure value may be welcomed to obtain a perfect fit with the theoretical calculations. Be that as it may, we would like to note that Fig. 7(d) is an enlarged view around the plateau, and the discrepancy between the experimental data and theory is already less than ~ 2 percent for the magnetization (and, hopefully, also for the critical magnetic fields) in the scale of the saturation values.

We added the above explanations with some more details in the newly-added “Supplementary Note 2” in the Supplementary information file. In addition, we added brief comments also in the main text (please see lines 301-311).

We hope that the above-mentioned additions of the detailed discussions provide solid justification of the applicability of our proposed model and the comparison of the model predictions with the previously published experimental results. We believe that the quality of the paper has been further improved, thanks to the constructive review.

4 Response to Reviewer #3

Reviewer: “One of the barriers that must be overcome in quantum computing, quantum transmission, etc., is the control of quantum spin and classical spin. This is an excellent paper

that contains the important information needed for this and attempts to clarify it entirely from experimental facts. Figure 1 in particular is educational and informative for many concerned researchers. Therefore, I recommend to submit.”

We thank Reviewer #3 for his/her positive comments and for recommending our paper for publication.

Reviewers' Comments:

Reviewer #1:

Remarks to the Author:

The revised version of the manuscript contains notable improvements and clarifications which make it well suited for publication. The authors have addressed all my questions adequately, and also answered all questions of other referees.

However, upon further studying the subject I came across the paper by Hosoi, Matsuura, and Ogata, J. Phys. Soc. Jpn. 87, 075001 (2018) (Ref.54 of the current manuscript), which theoretically analyzes the problem of CsCuCl₃ in a similar yet different way - via large-S expansion but without the "squeezing" and the mapping to the 2d employed in the current manuscript. Their findings, summarized in Figs.2 and 3, appear to be quite consistent with the present work. I, therefore, think that a critical comparison between the current theory and that of Ref.54 is required. Clearly, the present manuscript contains detailed comparison with the new experimental data which strengthens the scientific case put forward by the authors. Yet it would be nice to have a brief comparison of the two theoretical techniques used in these two works.

Reviewer #2:

Remarks to the Author:

The authors have clearly given very careful consideration to the comments on the previous version of this submission and I thank them sincerely for their constructive and helpful clarifications. I am very happy to recommend the revised manuscript for publication.

Reviewer #3:

Remarks to the Author:

This is an excellent paper that contains the important information needed for this and attempts to clarify it entirely from experimental facts. Figure 1 in particular is educational and informative for many concerned researchers. Therefore, I recommend to submit.

1 Summary of changes

1. We added a new paragraph to address the suggestion by Reviewer #1 at the end of the Results of the revised manuscript (highlighted in red).
2. We added in the Acknowledgements one more Grant Number “17H01142 (H.T.)”, which we have forgotten to include in the previous version (highlighted in red).
3. The section headings were revised (following the guidance).
4. The style of the texts in the figures was revised (following the guidance).
5. “Competing Interests” section was added (highlighted in red).
6. The order of the first and second affiliations of the first author was exchanged (highlighted in red).

2 Response to Reviewer #1

Reviewer: “The revised version of the manuscript contains notable improvements and clarifications which make it well suited for publication. The authors have addressed all my questions adequately, and also answered all questions of other referees. However, upon further studying the subject I came across the paper by Hosoi, Matsuura, and Ogata, *J. Phys. Soc. Jpn.* 87, 075001 (2018) (Ref. 54 of the current manuscript), which theoretically analyzes the problem of CsCuCl_3 in a similar yet different way – via large- S expansion but without the “squeezing” and the mapping to the 2d employed in the current manuscript. Their findings, summarized in Figs. 2 and 3, appear to be quite consistent with the present work. I, therefore, think that a critical comparison between the current theory and that of Ref. 54 is required. Clearly, the present manuscript contains detailed comparison with the new experimental data which strengthens the scientific case put forward by the authors. Yet it would be nice to have a brief comparison of the two theoretical techniques used in these two works.”

Response: We thank Reviewer #1 for the suggestion. In the analysis of Ref. 54, the intrachain coupling J_0 was assumed to be constant with the applied pressure, and the plateau formation was explained as resulting from the reduction of the effective anisotropy Δ . Our present analysis based on the parameter fittings with the experimental data has revealed that the change in Δ is not enough to explain the emergence of the plateau, but the enhancement of the quantum effects associated with the strong reduction of J_0 plays a key role for the pressure-induced classical-quantum crossover discussed in the Discussions.

We added one paragraph for the suggested comparison of our theoretical analysis with that of the previous study in Ref. 54 at the end of the Results of the revised manuscript. Please see the part highlighted in red.

3 Comment by Reviewer #2

Reviewer: “The authors have clearly given very careful consideration to the comments on the previous version of this submission and I thank them sincerely for their constructive and

helpful clarifications. I am very happy to recommend the revised manuscript for publication.”

4 Comment by Reviewer #3

Reviewer: “This is an excellent paper that contains the important information needed for this and attempts to clarify it entirely from experimental facts. Figure 1 in particular is educational and informative for many concerned researchers. Therefore, I recommend to submit.”